# Assessment of Torsional Amplification of Drift Demand in a Building Employing Site-Specific Response Spectra and Accelerograms

Yao Hu [1,*], Prashidha Khatiwada [1], Elisa Lumantarna [1] and Hing Ho Tsang [2]

1   Department of Infrastructure Engineering, The University of Melbourne, Melbourne, VIC 3010, Australia
2   School of Engineering, Swinburne University of Technology, Melbourne, VIC 3122, Australia
*   Correspondence: yaoh4@student.unimelb.edu.au; Tel.: +61-431-019-993

**Abstract:** This paper aims at giving structural designers guidance on how to transform seismic demand on a building structure from two-dimensional (2D) to three-dimensional (3D) in an expedient manner, taking into account amplification of the torsional actions. This paper is to be read in conjunction with either paper #3 or #4. Torsional amplification of the drift demand in a building is of major concern in the structural design for countering seismic actions on the building. Code-based seismic design procedures based on elastic analyses may understate torsional actions in a plan of asymmetric building. This is because the inability of elastic analyses to capture the abrupt increase in the torsional action as the limit of yield of the supporting structural walls is surpassed. Nonlinear dynamic analysis can provide accurate assessment of torsional actions in a building which has been excited to respond in the inelastic range. However, a 3D whole building analysis of a multi-storey building can be costly and challenging, and hence not suited to day-to-day structural design. To simplify the analysis and reduce the scale of the computation, closed-form expressions are introduced in this paper for estimation of the $\Delta_{3D}/\Delta_{2D}$ drift demand ratio for elastic conditions when buildings are subjected to moderate-intensity ground shaking. The drift demand of the 3D model can be estimated as a product of the 2D drift demand and the $\Delta_{3D}/\Delta_{2D}$ drift demand ratio. In dealing with higher-intensity ground shaking causing yielding to occur, a macroscopic modelling methodology may be employed. The estimated $\Delta_{3D}/\Delta_{2D}$ drift demand ratio of an equivalent single-storey building is combined with separate analysis for determination of the 2D drift demand. The deflection profile of the multi-storey prototype taking into account 3D effects, including torsional actions, is hence obtained. The accuracy of the presented methodologies has been verified by case studies in which drift estimates generated by the proposed calculation procedure were compared against results from whole building analyses, employing a well-established computer software.

**Keywords:** site-specific structural analysis; 3D nonlinear time history analysis; rapid nonlinear dynamic method; torsional amplification

## 1. Introduction

Irregular buildings in plan or elevation are common to fulfil architectural requirements [1]. Excessive structural drift demand can be experienced by the building in seismic conditions should there be asymmetry in the distribution of mass, stiffness, or strength in the lateral resisting elements, including the structural walls [2–4]. Severe damage to buildings and structural failure that were aggravated by asymmetry have been experienced in past earthquake events [5]. Structural analyses based on linear elastic behaviour are the common (and default) type of analyses employed in the design of most code-compliant buildings. The torsional actions of the buildings in post-yield conditions can be understated by this simplified form of analyses in spite of the structural design being code-compliant.

There are equivalent static analysis provisions stipulated by major design codes of practices (namely EC8 and ASCE/SEI 41-06 [6,7]) that are intended to allow for the effects

of torsional actions in an irregular building. These provisions have been found to be very conservative [8,9]. The nonlinear static analysis procedure (which is also known as pushover analysis) is well-publicised and has been progressively extended to allow for seismically induced torsional actions in the building [10,11]. Other analytical procedures have been developed to facilitate the uptake of structural design by circumventing the need of time history analyses of the whole building, whilst having the appeal of taking into account the effects of the higher modes and torsional actions. These techniques are namely multi-mode pushover (MMP) analysis [12], modal pushover analysis (MPA) [13,14], modified modal pushover analysis (MMPA) [15], consecutive modal pushover (CMP) [16], and the extended N2 method [17–19]. Whilst pushover analysis was originally meant to replace dynamic analyses with static analyses of the buildings, some advanced versions of such "approximate" methods might involve nonlinear time history analysis, forming part of the procedure. The original appeal of pushover analysis is, therefore, not retained in the various modified version. Consequently, despite being well-publicised, the uptake of the aforementioned modelling ("pushover analysis") techniques has been limited in practical design, and more so in regions of low to moderate seismicity.

Nonlinear time history analysis (NLTHA) is a more direct (and rigorous) method for assessment of the seismic drift demand of multi-storey building models that are excited beyond the post-yield limit [20,21]. The torsional response behaviour of a building featuring asymmetry and its sensitivity to changes in ground motion properties such as intensity, spectral frequency, and duration can be analysed by employing NLTHA [22]. However, this form of analysis may require intensive computational resources and can be very time-consuming if applied to the full model of a multi-storey building. There is potential for developing more efficient NLTHA methods for fast prediction of torsion actions in the inelastic range with a reasonable degree of accuracy. This paper was motivated by this potential when we set out to write it in contribution to the special issue. Methodologies introduced in this paper for dealing with nonlinear behaviour employ both the pushover analysis and NLTHA procedures.

The objective of this paper is to provide practical alternatives to design engineers to achieve fast estimates of the drift demand in a building that features asymmetry in both the linear and nonlinear ranges. Fast estimates of the drift demand in support of the design process enable the structural designer to anticipate any potential challenges throughout the design process and to optimise the design of the building for achieving satisfactory performance. An image scanning technique, as presented in Section 2, was used to assist a speedy identification of torsional parameters, namely eccentricity ($e_r$), elastic radius ($b_r$), and mass-radius of gyration ($r$), for input into the elastic drift assessment based on information presented on the floor plan of the buildings. Closed-form expressions and a solution technique that can be used for accurate determination of the $\Delta_{3D}/\Delta_{2D}$ drift demand ratio in elastic conditions are introduced in Section 3. The rest of the paper deals with the estimate of the drift demand in nonlinear inelastic conditions. Two routines are introduced in Section 4 for the calculation of the inelastic drift demand (represented in terms of the $\Delta_{3D}/\Delta_{2D}$ drift demand ratio) by employing the inelastic response spectrum and NLTHA of an equivalent single-storey building. The drift demand in the building can be found as the product of the $\Delta_{3D}/\Delta_{2D}$ drift demand ratio and the deflection of the building in two dimensions, as obtained from the analysis procedures introduced in [23,24]. Finally, a case study of a multi-storey building featuring asymmetry is presented in Section 5 to illustrate the use of the presented analytical procedures for given input information, such as site-specific accelerograms, or site-specific response spectra.

## 2. Image Scanning Method for Determining Torsional Parameters

Parameters that are specifically relevant to the assessment of torsional actions in a building include the eccentricity ratio ($e_r$) and the elastic radius ratio ($b_r$) [25]. The values of these parameters are not readily available, and their determination based on information presented on the floor plan of the building can be time-consuming. A method of image

scanning based on the use of OpenCV [26] employs the equivalent frame pair modelling technique to determine the position of the centre of mass (CM) and centre of rigidity (CR) of the building, the radius of gyration of mass ($r$), the value of $b_r$, which can be found by taking the square root of the ratio of torsional stiffness to translational stiffness of the building (normalised with respect to $r$), and the eccentricities '$e_{xr}$' and '$e_{yr}$', which is the offset between the CM and CR in the x and y direction, respectively (normalised with respect to $r$). The analyses as described involve the use of an image scanning technique and are automated via purposely built software. Figure 1 presents the case study example of a Y-shaped building for illustration of the use of the facility to determine the location of the CM and CR, along with the value of r, and that of $e_{xr}$, $e_{yr}$, and $b_r$.

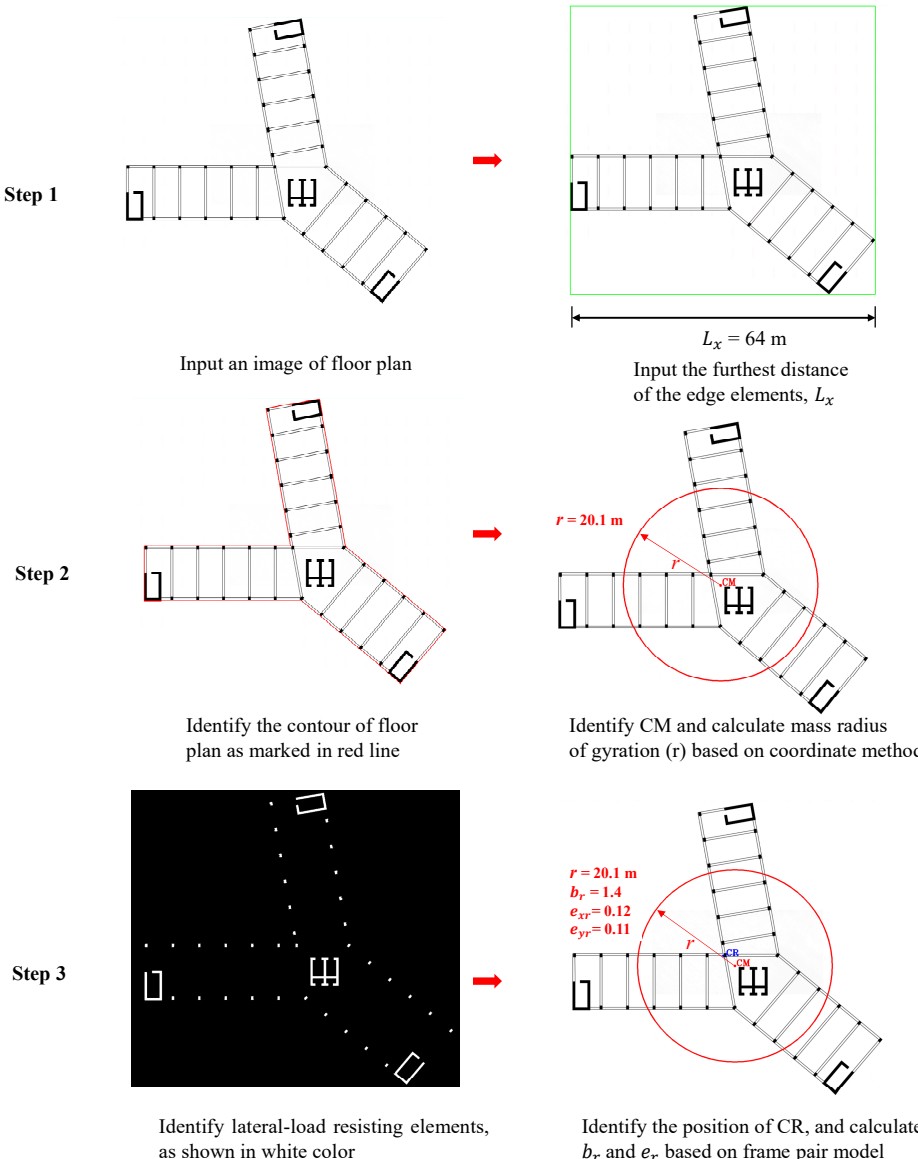

**Figure 1.** Schematic view of torsional stability assessment using the image scanning tool.

The automation is summarised in three steps, as outlined in the following:

**Step 1:** The users are required to scan the floor plan of the building for input into the software and to key in the value of $L_x$, which is the distance between the considered edge element and the position of the CM.

**Step 2**: The "contours" of the floor plan are found by use of OpenCV. The location of the CM and the value of $r$ can be found using the coordinate method [3].

**Step 3:** Lateral load-resisting elements such as walls and columns are then identified, along with the location of the CR. The values of $e_r$ and $b_r$ can be calculated using the frame pair modelling technique [27].

With the case study building featuring a Y-shaped floor plan (Figure 1), the values returned by the purposely built software (involving image scanning) for estimation of $r$, $b_r$, $e_{xr}$, and $e_{yr}$ were 20.1 m, 1.4, 0.12, and 0.11 (in comparison with 20 m, 1.5, 0.15, and 0.12, which were obtained from static analysis using the program SPACEGASS, as shown in Figure A1). The computational efficiency and accuracy of the image scanning have been verified. The torsional parameters so obtained were next employed for predicting the $\Delta_{3D}/\Delta_{2D}$ drift demand ratio in the elastic state. Details can be found in Section 3.

## 3. Site-Specific 3D Linear Elastic Dynamic Analysis

The dynamic response of the building to site-specific soil-surface motion can be determined as the product of the $\Delta_{3D}/\Delta_{2D}$ drift demand ratio and the 2D displacement demand, which can be found by 2D dynamic modal analysis of the building along with the use of the elastic response spectrum. As shown in Figure 2, $\Delta_{3D}$ refers to the maximum displacement at one of the edges of the single-storey model (i.e., $\Delta_{stiff}$ and $\Delta_{flexible}$), whereas $\Delta_{2D}$ refers to the maximum displacement of the equivalent single-degree of freedom (SDOF) model, $B_x$ is the distance of the edge element from the centre of mass, and $K_y$ and $K_x$ are the stiffness of structural walls in the perpendicular and orthogonal directions. The $\Delta_{3D}/\Delta_{2D}$ drift demand ratio can be found from the dynamic analysis of the equivalent single-storey model for analysing its elastic torsional coupling. Modal combinations were carried out in accordance with the square root sum of squares (SRSS) combination rule. The solution for the $\Delta_{3D}/\Delta_{2D}$ drift demand ratio is presented as closed-form expressions, Equations (1a)–(1c). Thus, the need to undertake dynamic analysis for modelling the effects of torsional coupling is spared. Derivation of the predicted relationships based on the principles as described can be found in [28,29].

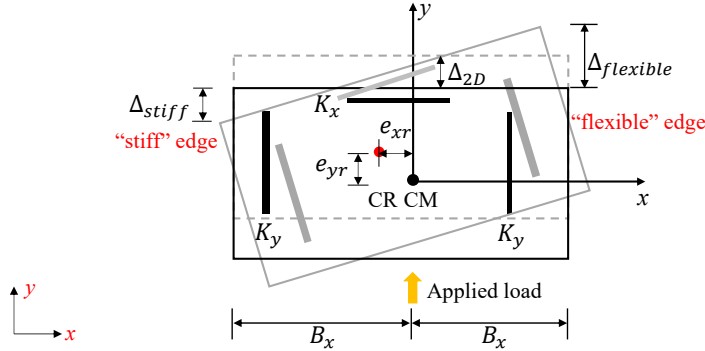

**Figure 2.** Single-storey building model featuring asymmetry subjected to y-directional spectral loads.

In dealing with uniaxial eccentricity, the following expressions may be used:
For the acceleration-controlled conditions,

$$\frac{\Delta_{3D}}{\Delta_{2D}} = \sqrt{\sum_{j=1}^{2}\left[\left(1+\theta_j(\pm B_{xr})\right)PF_j \cdot \frac{1}{\lambda_j^2}\right]^2} \tag{1a}$$

For the velocity-controlled conditions,

$$\frac{\Delta_{3D}}{\Delta_{2D}} = \sqrt{\sum_{j=1}^{2}\left[\left(1+\theta_j(\pm B_{xr})\right)PF_j \cdot \frac{1}{\lambda_j}\right]^2} \tag{1b}$$

For the displacement-controlled conditions,

$$\frac{\Delta_{3D}}{\Delta_{2D}} = \sqrt{\sum_{j=1}^{2} \left[ \left(1 + \theta_j(\pm B_{xr})\right) PF_j \right]^2} \tag{1c}$$

where $B_{xr} = \frac{B_x}{r}$, $\lambda_j = \frac{1+b_r^2+e_{xr}^2}{2} \pm \sqrt{\left(\frac{1-b_r^2-e_{xr}^2}{2}\right)^2 + e_{xr}^2}$, $\theta_j = \frac{\lambda_j^2-1}{e_{xr}}$, and $PF_j = \frac{1}{1+\theta_j^2}$. In dealing with biaxial eccentricity, let $a = k_x/k_y$, and the values of $\lambda_j$ and $PF_j$ can be obtained by solving Equation (2). Both MATLAB and EXCEL contain facilities to solve these types of problems.

$$Det \begin{bmatrix} a - \lambda_j^2 & 0 & ae_{yr} \\ 0 & 1 - \lambda_j^2 & e_{xr} \\ ae_{yr} & e_{xr} & ae_{yr}^2 + e_{xr}^2 + b_r^2 - \lambda_j^2 \end{bmatrix} = 0 \tag{2a}$$

$$PF_j = \frac{1}{1 + \left( \frac{e_{yr}}{e_{xr}} \left( \frac{a - a\lambda_j^2}{a - \lambda_j^2} \right) \right)^2 + \theta_j^2} \tag{2b}$$

The $\Delta_{3D}/\Delta_{2D}$ drift demand ratio is controlled by parameters $b_r$, $e_{xr}$, and $e_{yr}$, which can be readily found by the image scanning technique, as introduced in Section 2. Procedures for obtaining the response spectra and time histories in 2D are presented below.

**2D Linear Response Spectrum Analysis**

**Step 1:** Generate six pairs of site-specific soil-surface elastic acceleration response spectra for each of the reference periods, 0.2 s, 0.5 s, 1 s, and 2 s, using the procedure described by Hu et al. [30,31]. The accelerogram generation facility is available for free online via the website 'quakeadvice.org', which was developed and currently managed by the authors and co-workers. Determine the mean elastic acceleration response spectrum '$RSA(T_n)$' for each reference period group.

**Step 2:** Determine the natural period of vibration of the building along with the displacement coefficients covering multiple modes of vibration of the structure using either the eigen solver, that is available in many commercial packages, or the simplified procedure described in [32]. Calculate the maximum displacement of the equivalent SDOF model corresponding to mode "$j$", which has a natural period $T_n = T_{n,j}$. Transform the SDOF modal displacements and have them combined (using the SRSS modal combination rule) to form an estimate of the deflection of a multi-degree of freedom (MDOF) model based on the given displacement coefficients. The design 2D displacement is the maximum 2D displacement obtained from the most critical response spectrum (out of the four reference period groups).

**2D Linear Time History Analysis**

**Step 1**: Generate 12 to 16 soil-surface accelerograms (2 to 6 from each of the 4 reference period groups) following the procedure proposed by Hu et al. [30,31], which has been implemented in the online tool available at 'quakeadvice.org'.

**Step 2**: Determine the angular velocity, $\omega_j$, and displacement coefficient, $P_j$, of the building using an eigen analysis or the simplified procedure described in [32].

**Step 3**: Using Equation (A2) and the information from Step 2, determine the elastic time history displacement response for different modes of vibration.

**Step 4**: For each mode of vibration, multiply the SDOF responses with the modal displacement coefficient obtained from Step 2 to calculate the displacement time history in the floor levels of the building. Additionally, determine the peak displacement for each accelerogram.

**Step 5**: Apply the modal combination rule to determine the combined floor-level (MDOF) displacement response. Linear addition for combining the time history response and the square root sum of squares for combining the peak displacement response is

recommended. The 2D design displacement is equal to the maximum of the mean of the peak displacement response obtained for each of the four reference period groups.

## 4. Site-Specific 3D Nonlinear Dynamic Analysis

*4.1. Determination of the Maximum Inelastic $\Delta_{3D}/\Delta_{2D}$ Drift Demand Ratio*

The $\Delta_{3D}/\Delta_{2D}$ drift demand ratio in inelastic conditions can be calculated using two computational routines. Routine 1 is for undertaking pushover analysis, also involving nonlinear response spectrum analysis (NRSA), whereas Routine 2 is for rapid nonlinear time history analysis in 3D (RNLTHA-3D). Each of these routines are described in detail below. Readers can make their own choice between the two alternative modelling approaches.

**Routine 1: Inelastic $\Delta_{3D}/\Delta_{2D}$ by pushover analysis and nonlinear response spectrum analysis**

Pushover analysis is a simplified form of nonlinear static analysis for the seismic assessment of a structure [33,34]. When performing pushover analysis of a building featuring plan asymmetry, the two issues that need to be considered are: (i) the effects of stiffness degradation and changes in dynamic properties in inelastic conditions, and (ii) contribution of the higher modes that account for a coupled lateral–rotational response [35]. In this section, the drift demand ratio ($\Delta_{3D}/\Delta_{2D}$) is discussed, which is the ratio of the seismic demand as derived from the analysis of an equivalent single-storey model with three degrees of freedom, to the translational drift demand, as derived from the analysis of an equivalent SDOF model.

The equivalent stiffness of a single-storey model at every stage of the pushover analysis is defined as the secant stiffness, taking into account stiffness degradation. The pushover analysis procedure for the calculation of the $\Delta_{3D}/\Delta_{2D}$ drift demand ratio consists of the following steps, which are diagrammatically presented in Figure 3.

**Step 1**. Apply pushover analysis to a 3D and a 2D multi-storey building, separately.

**Step 2**. Transform a 3D multi-storey building into an equivalent single-storey model with three degrees of freedom. The corresponding 2D multi-storey building model is transformed into an equivalent SDOF model using Equations (A1a)–(A1c), presented in Appendix B.

**Step 3**. Calculate torsional parameters: $e_r$ and $b_r$, and the natural period ($T_n$) for each loading stage in Step 1 using Equations (A1d)–(A1f), presented in Appendix B, assuming that the equivalent stiffness of the single-storey model at each nonlinear stage is defined by their respective secant stiffness. The mass radius of gyration of a floor plan with irregular geometry can be identified using the image scanning method, as presented in Section 2.

**Step 4**. Calculate the maximum $\Delta_{3D}/\Delta_{2D}$ drift demand ratio using Equations (1a)–(1c) for each loading stage in Step 1, and plot a graph showing $\Delta_{2D}$ of a SDOF model (*x*-axis of the plot) against $\Delta_{3D}/\Delta_{2D}$ (*y*-axis of the plot).

**Step 5**. Estimate the 3D drift demand as the product of $\Delta_{3D}/\Delta_{2D}$ and the 2D drift demand as estimated from the method explained in Section 4.2.

Note, the procedure presented in the foregoing is based on the assumption of uniaxial asymmetry (i.e., $e_{yr} = 0$) due to the limitations of Equations (1a)–(1c). When dealing with a building featuring biaxial asymmetry, predictions given by the procedure are expected to err on the safe side.

**Routine 2: Maximum inelastic $\Delta_{3D}/\Delta_{2D}$ by nonlinear time history analysis**

Nonlinear time history analysis is perceived to be more rigorous and time-consuming than pushover analysis, which has been used to simplify a dynamic analysis into a static analysis. Similarly, analytical procedures employing the response spectrum have also been used to simplify a nonlinear problem into a linear problem. The supposedly simplified procedures can be made rather complex in order to achieve good accuracy. Multiple versions of these simplified procedures can also present challenges to the users.

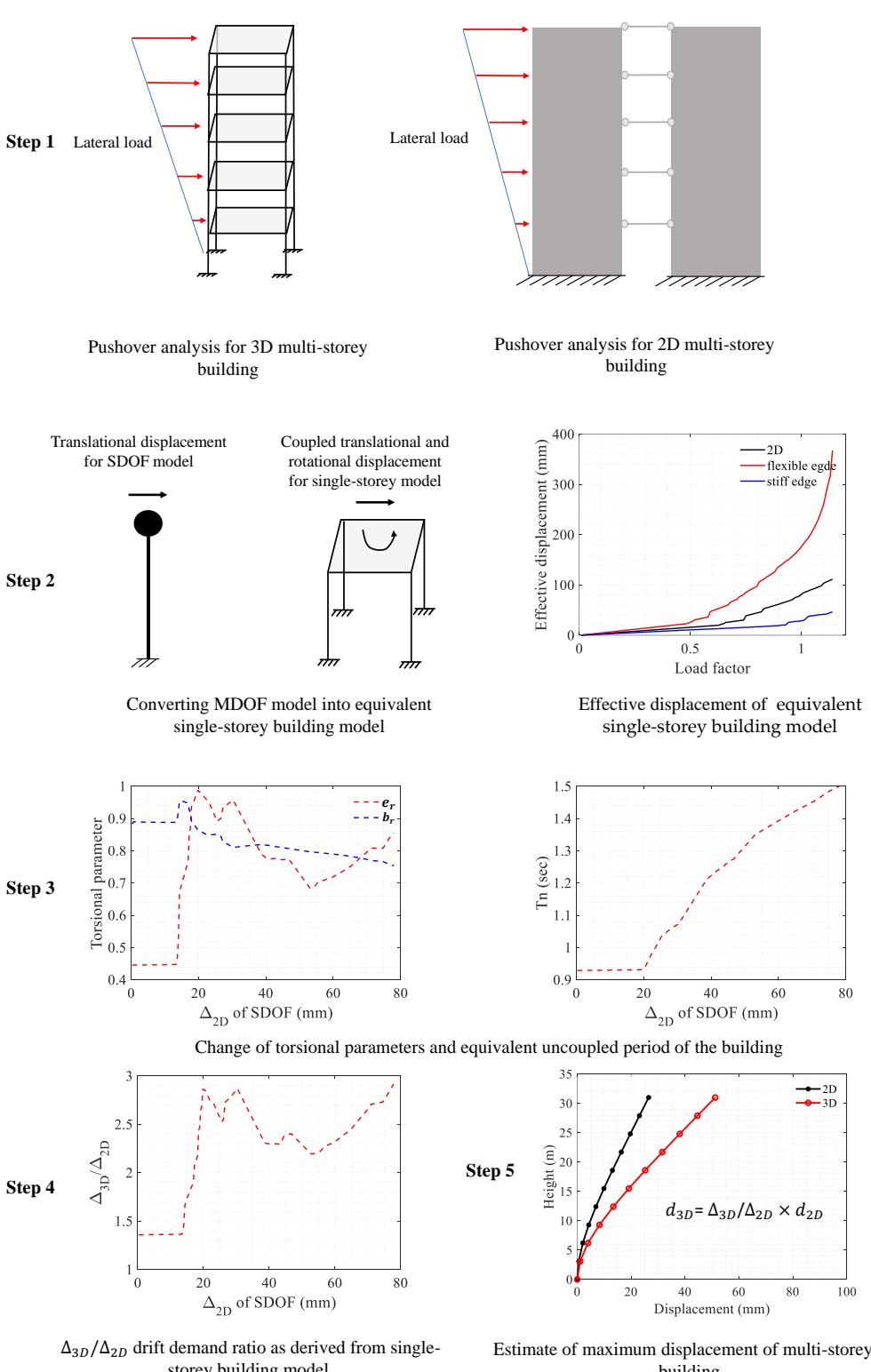

**Figure 3.** The procedure for obtaining the $\Delta_{3D}/\Delta_{2D}$ drift demand ratio vs. 2D nonlinear displacement using nonlinear response spectrum analysis.

The appeal of a method depends not only on the computational time but also on its reliability and conceptual simplicity. The computational time of a nonlinear time history analysis (NLTHA) depends on the nature of the analysis and the number of degrees of freedom involved in the modelling. In this section, the procedure for conducting rapid

nonlinear time history analysis (RNLTHA) for the determination of 3D drift demand is presented (Figure 4).

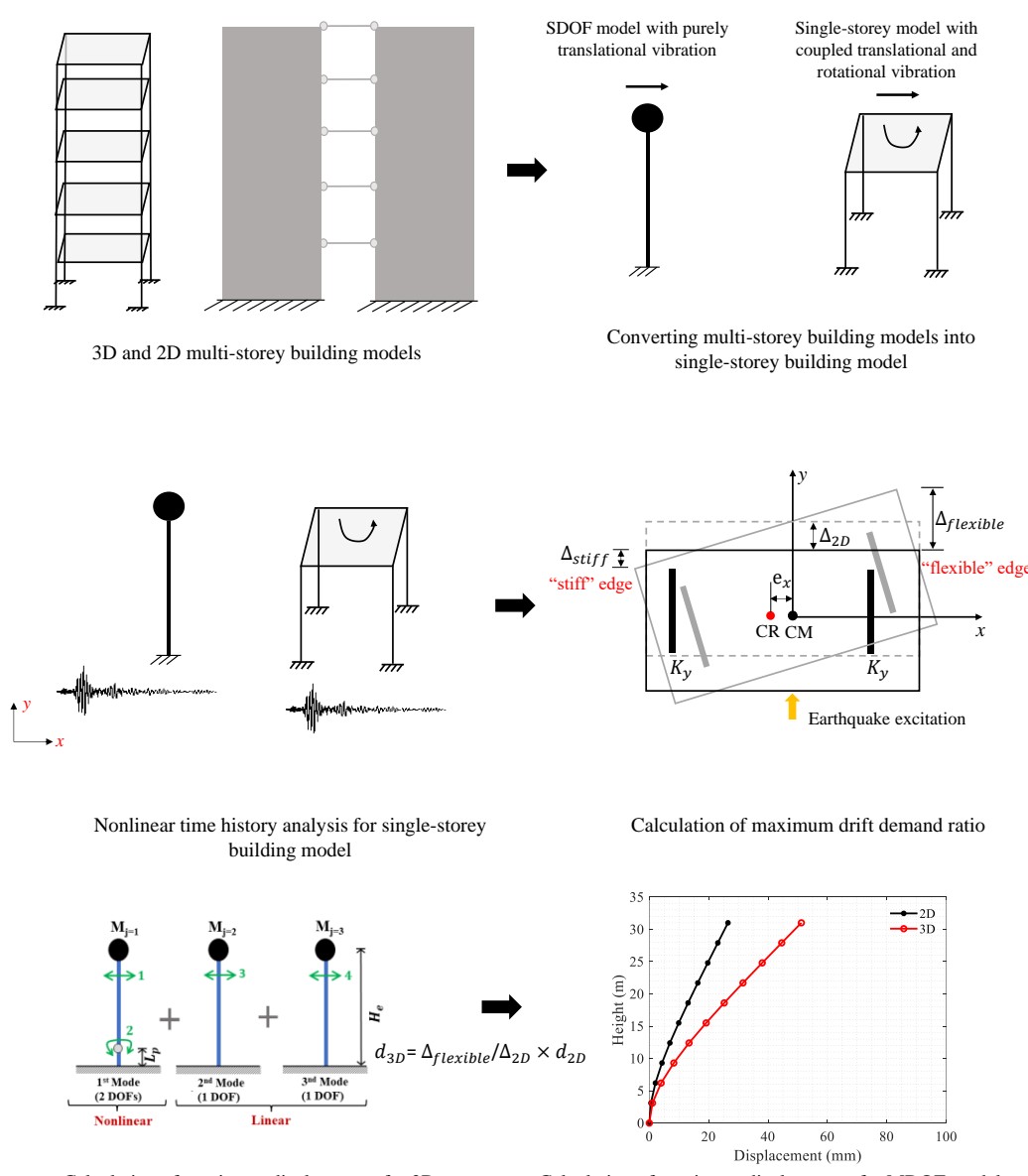

**Figure 4.** The procedure of obtaining the $\Delta_{3D}/\Delta_{2D}$ drift demand ratio vs. nonlinear displacement using rapid nonlinear time history analysis.

**Step 1**: Transform a 3D whole model of a building into an equivalent single-storey model and an equivalent SDOF model, respectively. The effective mass and height of the equivalent single-storey model are calculated using: $M_e = (\sum m_i h_i)^2 / \sum m_i h_i^2$, and $H_e = \sum m_i h_i^2 / \sum m_i h_i$, assuming that the distribution of mass $(m_i)$ up the height of the building $(h_i)$ is uniform and that the deflection shape takes the form of an inverted triangle (which is a reasonable assumption to make for a building with uniform distribution of mass up to the building height). Alternatively, the equivalent mass of the building can be approximated as 0.7 times the total weight $(M_{total})$.

**Step 2**: Conduct nonlinear time history analysis to determine the maximum displacement demand of both the single-storey model $(\Delta_{3D})$ and that of the SDOF model $(\Delta_{2D})$. The $\Delta_{3D}/\Delta_{2D}$ drift demand ratio can be calculated for each individual accelerogram employed in the analysis.

**Step 3**: The 3D drift demand is calculated as the product of the $\Delta_{3D}/\Delta_{2D}$ drift demand ratio and the 2D drift demand, as derived from the analysis employing the procedure introduced in [23,24] (Figure A2), as described in detail in Section 4.3.

*4.2. Nonlinear Response Spectrum Analysis*

Nonlinear response spectrum analysis is in alignment with Routine 1, as introduced in Section 4.1. The 3D displacement response of the building found on the soil surface can be determined as the product of the $\Delta_{3D}/\Delta_{2D}$ drift demand ratio and the deflection of the 2D model of the building. The drift demand ratio can be obtained from calculations based on the method described in Section 4.1, and the deflection of the building in 2D can be found by employing nonlinear response spectrum analysis [23,24]. The procedure is summarised below.

**Step 1:** Generate six pairs of site-specific soil-surface elastic acceleration response spectra for each of the reference periods: 0.2 s, 0.5 s, 1 s, and 2 s, using the mean elastic acceleration response spectrum '$RSA_{elastic}$' for each reference period group.

**Step 2:** Construct the inelastic response spectrum of the seismic actions in the acceleration–displacement response spectrum (ADRS) format based on plotting $RSD_{inelastic}$ vs. $RSA_{inelastic}$ using Equation (3) [36]. An alternative method such as that proposed by Chopra and Goel [37] may be employed.

$$RSA_{inelastic} = \frac{RSA_{elastic}}{\min\left[(\mu - 1)\frac{T_n}{T_c} + 1, \mu\right]\Omega} \tag{3a}$$

$$RSD_{inelastic} = \frac{\mu}{\min\left[(\mu - 1)\frac{T_n}{T_c} + 1, \mu\right]} RSA_{elastic} \left(\frac{T}{2\pi}\right)^2 \tag{3b}$$

where the ductility ratio, $\mu$, is defined as the ratio of maximum displacement ($\Delta_u$) to the corresponding displacement at the onset of yielding ($\Delta_y$), the overstrength factor $\Omega = 9.1n^2 - 3.6n + 1.6$ (where $n$ is the axial load ratio of the critical structural wall), $T_n$ is the natural period of vibration of the structure, $T_c$ is the first corner period of the response spectrum, and $T$ is the time domain of the response spectrum.

**Step 3:** Construct the inelastic force–displacement curve of the form shown in Figure 5 using a pushover analysis procedure which involves sectional analysis of the structural walls [38–40]. Divide the forces by the modal mass of the equivalent SDOF model in order for the force–displacement curve to be transformed into an acceleration–displacement curve, where $I_{gross}$ is the gross second moment of inertia of the wall cross-section, and $E_c$ is the Young's modulus of concrete. Means of determining the first crack, $\phi_{cr}$, yield curvature, $\phi_y$, ultimate curvature, $\phi_u$, the effective second moment of area of the structural elements, $I_{eff}$, plastic hinge length, $L_p$, and yield penetration, $L_{sp}$, can be found in the literature [23,24].

**Step 4:** Overlay the inelastic response spectral curve (representing the seismic demand) as obtained in Step 2 onto the pushover curve as obtained in Step 3 for identifying the point of intercept, which is also known as the performance point, showing both the design acceleration and displacement demand of the equivalent SDOF model, which is denoted herein as $\Delta_{design,SDOF}$.

**Step 5:** Make use of $\Delta_{design,SDOF}$ obtained from Step 4 to infer the displacement demand ($\Delta_{MDOF,i}$) at the floor levels (of height $h_i$) of the MDOF model using Equation (4):

$$\Delta_{MDOF,i} = 1.5\,\Delta_y\left(\frac{h_i^2}{H_e^2} - \frac{h_i^3}{3H_e^3}\right) + \left(\Delta_{design,SDOF} - \Delta_y\right) \times \left(\frac{h_i - 0.5L_p + L_{sp}}{H_e - 0.5L_p + L_{sp}}\right) \tag{4}$$

where $\Delta_y$ is the yield displacement, $h_i$ is the height of storey '$i$' aboveground, $H_e$ is the effective height, and $\Delta_{design,SDOF}$ is the design displacement obtained from Step 4.

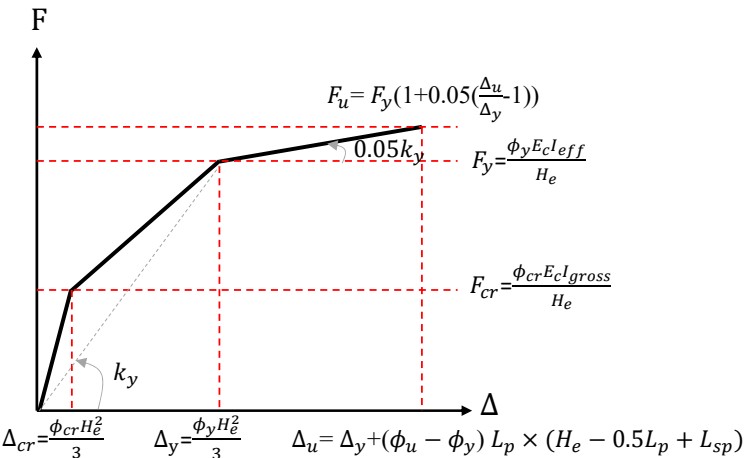

**Figure 5.** Trilinear pushover curve.

*4.3. Nonlinear Time History Analysis*

Nonlinear time history analysis is in alignment with Routine 2 in Section 4.1. The 3D time history displacement response of the building in response to the excitation caused, as defined by the soil-surface accelerograms, is determined by multiplying the $\Delta_{3D}/\Delta_{2D}$ drift demand ratio, as obtained from Routine 2 in Section 4.1, with the time history response of the displacement, as obtained from the analysis procedures introduced in [23,24]. The rapid nonlinear time history analysis 2D (RNLTHA-2D) procedure is summarized below:

**Step 1**: Convert a multi-storey building model into an equivalent SDOF model. The modal lumped masses, $M_j$, angular velocity, $\omega_j$, and displacement coefficient, $P_j$, of the SDOF model are calculated using the dynamic modal properties of the building or by employing a simplified procedure, as provided in [32].

**Step 2**: Generate 12 to 16 soil-surface accelerograms (2 to 6 from each of the 4 reference periods: 0.2 s, 0.5 s, 1 s, and 2 s) using the procedure described by Hu et al. [30,31], which is available for free online at 'quakeadvice.org'.

**Step 3**: Determine the displacement time history of SDOF elastic models corresponding to various modes of vibration using Equation (A2) to find the displacement of the next time step in a step-by-step integration procedure.

**Step 4:** For the first mode of vibration, determine the inelastic force ($F_N$) to the corresponding elastic displacement using the force–displacement curve of Figure 4 and the hysteresis model of Figure A3. Make use of the relationship along with Equation (A3) for finding the inelastic displacement of the next time step in a step-by-step integration procedure.

**Step 5**: For each mode of vibration, multiply the SDOF responses with the modal displacement coefficient as obtained from Step 1 to obtain the displacement time history at the floor levels up the height of the building. Determine the combined displacement time histories at the floor levels of the MDOF model based on taking the sum of the modal contributions.

**Step 6**: Repeat the analysis for each site-specific soil-surface accelerogram to determine the respective 2D MDOF displacement time histories of the building floors.

**Step 7:** Calculate the mean of the peak floor displacement for each of the four reference periods and take these displacements as the 2D design floor displacement demand.

**5. Case Study**

A 10-storey, 31 m-tall RC building, which is supported by shear and core walls, was used as the case study building for illustrating the application of the RNLTHA procedure for the calculation of the seismic displacement demand of a multi-storey building featuring asymmetry. Note that RNLTHA-3D is essentially the solution to RNLTHA-2D multiplied by the $\Delta_{3D}/\Delta_{2D}$ drift demand ratio. The case study building (with the floor plan and structural details shown in Figure 6) is located in the part of Australia (which is a stable

continental region) which has a hazard value of 0.08 g for a 2% probability of exceedance in 50 years. The shear walls (Walls 1 and 2), which provide lateral support to the building, have material properties, as listed in Table A1. The 3D numerical models of the case study building were derived using the program SeismoStruct Version 2021 [41]. Estimates of the modal masses, modal natural periods, and modal displacement coefficients are summarised in Table A2. The seismic floor lumped mass (dead load + 0.3 × imposed load) of 420 tonnes was estimated by considering an imposed load of 2 KPa, and a superimposed dead load of 1 KPa and facade load of 1 KPa.

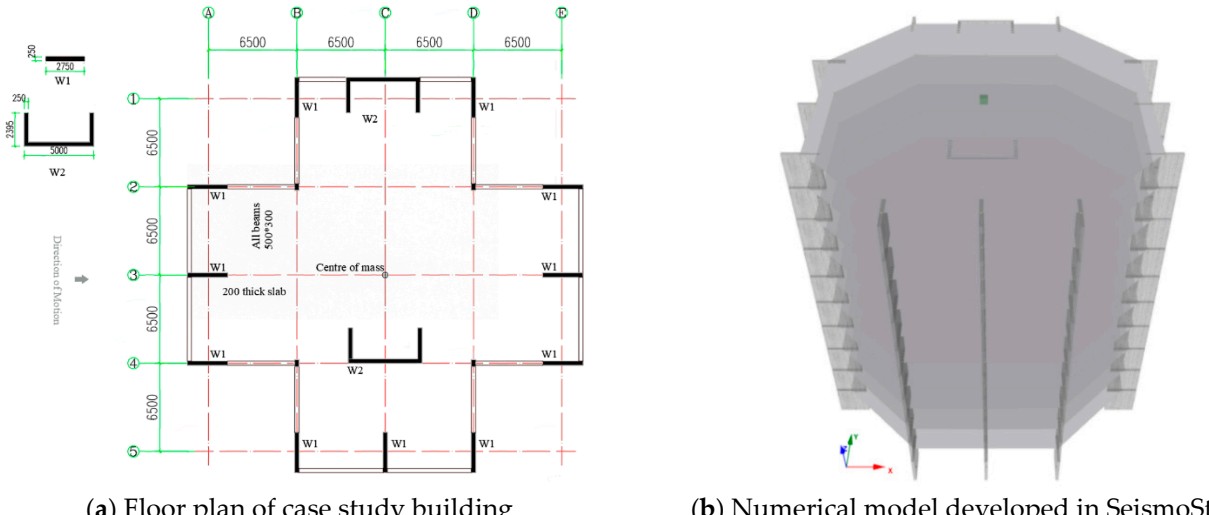

(**a**) Floor plan of case study building      (**b**) Numerical model developed in SeismoStruct

**Figure 6.** Floor plan of case study building and numerical models.

The online tools available at "quakeadvice.org" [42] were used to source and select the twenty-four code-compliant earthquake accelerograms on bedrock, which were sourced from the PEER database (refer Table A3 for the listing) [23]. A reverse/oblique fault, magnitude range of ±0.3 $M_w$, Joyner–Boore distance range of ±30 km, and $V_{S,30}$ of 1000 m/s were used to source the earthquake records representing bedrock excitations. A suite of twenty-four soil-surface elastic response spectra (with six spectra for each of the four reference periods) and fourteen soil-surface accelerograms (two for 0.2 s, four for 0.5 s, six for 1 s, and two for 2 s reference periods) were generated by site response analyses of soil column models using the online tools. The borelog presented in [43] provides details of a soil column having a site natural period of 0.61 s (representing site class D as per AS1170.4-2007) and was used as the example soil column [44]. The mean soil-surface response spectrum for each of the four reference periods is presented in Figure A3. Time histories of the soil-surface accelerograms are presented in Figure A4.

The four elastic mean site-specific acceleration response spectra (representing the four reference periods) have been transformed into the inelastic acceleration displacement response spectrum (ADRS) format using Equation (3). These inelastic ADRS curves are overlaid on top of the pushover capacity curve, which has been simplified into the tri-linear form as shown in Figure 7 (green line) (refer to Figure 5 for the generic construct of the pushover curve model). The force capacity was calculated by adding the capacity of the six rectangular walls (W1) and the two C-shaped walls (W2), while the displacement capacity was determined by the wall having the lesser capacity (Wall 2). The "force" quantities have been transformed into "acceleration" quantities by dividing by the effective mass (which is approximately 0.7 times the total mass). The point of intercept of the demand and capacity curves yields an estimate of 128 mm, which is the 2D inelastic SDOF design displacement (refer to the red dot in Figure 7).

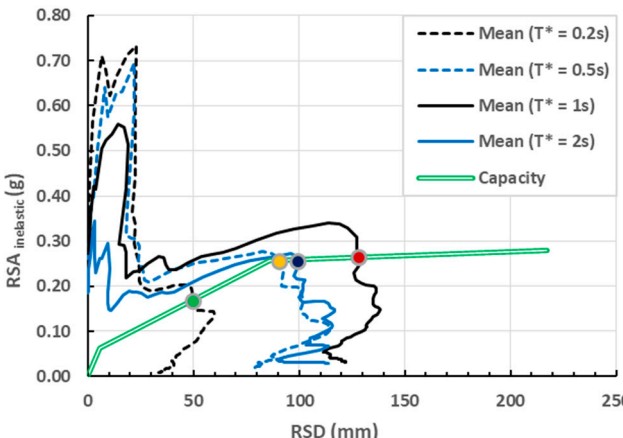

**Figure 7.** Superimposing of the mean inelastic ADRS demand and inelastic capacity curve.

By employing Routine 1, which was introduced in Section 4.1, the maximum inelastic drift demand of the 3D single-storey building normalised with respect to the 2D SDOF displacement demand is found, as shown in Figure 8a. As the 2D displacement demand of SDOF is estimated to be 128 mm, as shown in Figure 7, the drift demand ratio is calculated to be about 1.4, as observed in Figure 8a. The displacement demand of the SDOF model is then used for estimating the maximum deflection profile in 2D using Equations (3a) and (3b), as shown by the dark-coloured line in Figure 8b. The 3D displacement profile is obtained by the product of the 2D displacement profile and the $\Delta_{3D}/\Delta_{2D}$ ratio of 1.4. The displacement value of SDOF in Figure 8a (i.e., 128 mm) is approximately 0.65 times the displacement at the roof of the 2D multi-storey building (i.e., 196 mm) in Figure 8b based on eigenvalue analysis. The maximum displacement demand of the 3D building model is estimated to be 273 mm (being the product of the $\Delta_{3D}/\Delta_{2D}$ ratio of 1.4 and the maximum 2D displacement demand of 196 mm). Applying the amplification factor of 1.4 on the dark-coloured line affords the maximum deflection profile in 3D, as shown by the red-coloured line in the figure. Finally, the deflection profile in 3D is compared with predictions from SeismoStruct, as shown by the blue-coloured line in Figure 8b, to verify the accuracy of the simplified modelling technique.

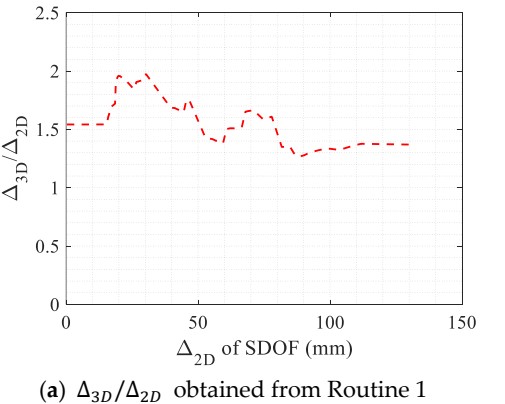

(**a**) $\Delta_{3D}/\Delta_{2D}$ obtained from Routine 1

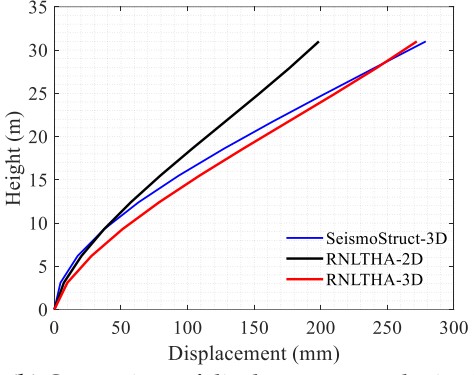

(**b**) Comparison of displacements as obtained from Routine 1 and SeismoStruct.

**Figure 8.** Comparison of the seismic demand as obtained from Routine 1 (RNLTHA-2D and 3D) with SeismoStruct-3D.

The proposed methodology shown in Routine 2 (see Figure 4) is investigated using the same case study building. The drift demand ratio of the single-storey building is calculated for each individual accelerogram. The 2D displacement of the building is calculated using the rapid nonlinear time history analysis 2D (RNLTHA-2D) procedure (ignoring torsion

amplification), and 3D displacement demand is derived by the product of the $\Delta_{3D}/\Delta_{2D}$ drift demand ratio and the 2D drift demand. The comparison of the maximum deflection estimated by rapid nonlinear time history analysis in 3D (RNLTHA-3D) (obtained following Routine 2 and the procedure presented in Section 4.3) against predictions by SeismoStruct has been extended to cover for earthquake excitations corresponding to the four reference periods, as shown in Figure 9 and Table 1. Comparisons of the peak displacement are shown for each accelerogram in Table A4. The comparison statistics show good agreement, as errors between the two sets of estimates were only up to 14.6% of the results predicted by SeismoStruct (the results shown in Figure 9 are generally more consistent when the reference period T* is equal to or more than 0.5 s). The proposed methodology has been shown to be able to predict the seismic demand of 3D buildings with reasonable accuracy. One more case study building with a different configuration of structural walls has also been investigated and the details can be found in Appendix H.

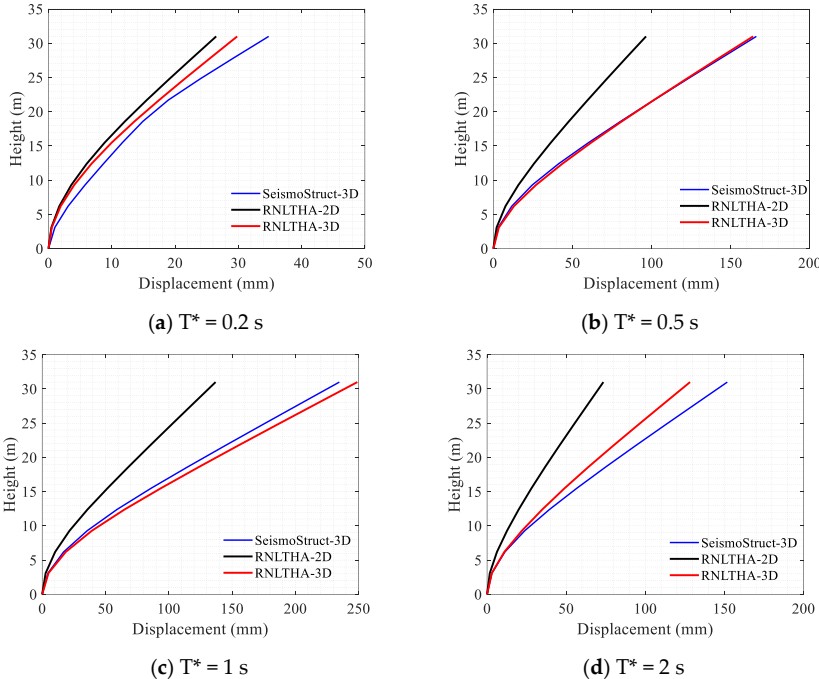

(**a**) T* = 0.2 s

(**b**) T* = 0.5 s

(**c**) T* = 1 s

(**d**) T* = 2 s

**Figure 9.** Comparison of the maximum displacement profile of each of the four reference periods obtained using Routine 2 (RNLTHA-3D) and SeismoStruct.

**Table 1.** Comparison of maximum 3D roof displacement obtained from the proposed RNLTHA-3D and SeismoStruct.

| Reference Period (T*) | $\Delta_{roof,\text{RNLTHA}-3D}$ (mm) | $\Delta_{roof,\text{SeismoStruct}}$ (mm) | Difference $\left(\frac{\Delta_{roof,\text{SeismoStruct}}-\Delta_{roof,\text{RNLTHA}-3D}}{\Delta_{roof,\text{SeismoStruct}}}\right)\times100\%$ |
|---|---|---|---|
| 0.2 s | 30 | 35 | 14.3% |
| 0.5 s | 162 | 166 | 2.4% |
| 1 s | 249 | 235 | −6.0% |
| 2 s | 129 | 152 | 14.6% |

The design roof displacement predicted from nonlinear time history analysis using Routine 2 is 249 mm, which is about 10% less than the value of 278 mm from nonlinear response spectrum analysis.

## 6. Conclusions

This paper aimed to facilitate the design of a multi-storey building for seismic safety by fast-tracking its dynamic analysis. Structural designers may then be able to exercise

better control of the design process. The fast-tracking can be achieved through waiving the need to subject the whole building to dynamic analyses, which can be very costly. The early part of this paper dealt with linear elastic analysis. The analysis was divided into two parts: (i) modelling the dynamic response behaviour of the building in 2D and (ii) modelling torsional actions by use of closed-form expressions to yield estimates of the $\Delta_{3D}/\Delta_{2D}$ drift demand ratio. To deal with inelastic behaviour, two approaches were introduced. The first approach made use of pushover analysis in conjunction with the inelastic response spectra for estimating inelastic drift demand. An equivalent single-storey building model was also developed for estimation of the torsional actions (Routine 1). The second approach involved subjecting macro-models of the building to nonlinear time history analyses for tracking the formation of the plastic hinge at the base and torsional rotation of the building (Routine 2). With both routines, the drift demand at the edge of the building was taken as the product of the $\Delta_{3D}/\Delta_{2D}$ drift demand ratio, as derived from the analysis of the equivalent single-storey building, and the displacement demand of the building in 2D. The application of the presented procedures was illustrated by a case study, which included a comparison of the drift demand derived from the newly introduced methods with results from whole building analysis employing an established software.

**Author Contributions:** Conceptualisation, E.L.; methodology, Y.H. and P.K.; software, Y.H. and P.K.; validation, E.L. and H.H.T.; resources, E.L.; writing—original draft preparation, Y.H. and P.K.; writing—review and editing, E.L.; supervision, E.L. and H.H.T. All authors have read and agreed to the published version of the manuscript.

**Funding:** This research received no external funding.

**Institutional Review Board Statement:** Not applicable.

**Informed Consent Statement:** Not applicable.

**Data Availability Statement:** Not applicable.

**Acknowledgments:** The authors acknowledge the financial support given by the University of Melbourne through its postgraduate research scholarship scheme. This work was also supported by the Commonwealth Australia through the Cooperative Research Centre program.

**Conflicts of Interest:** The authors declare no conflict of interest.

## Appendix A. Comparison of Torsional Parameters Obtained by Using the Imaging Scanning Method and the 3D Numerical Model

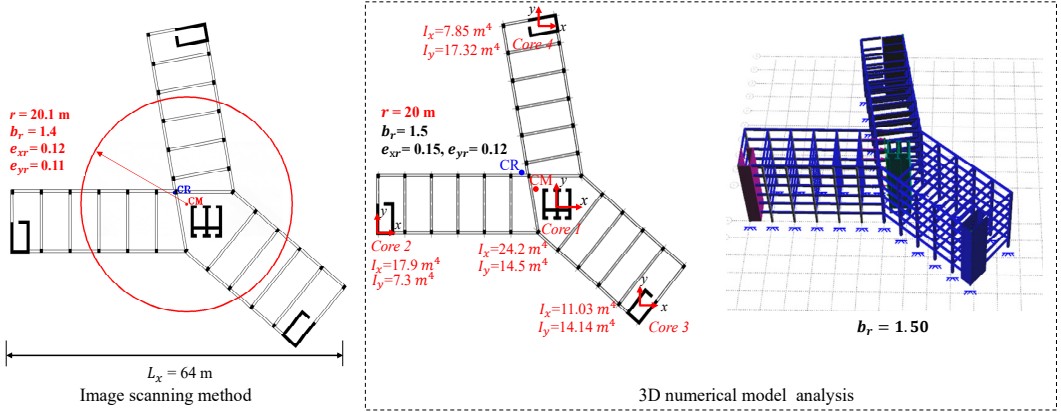

**Figure A1.** Comparisons of results obtained from the image scanning method and the 3D numerical model.

## Appendix B. Calculation of the Torsional Parameters from the Static Analysis Procedure

The 3D and 2D (3D model restrained for torsion/rotation) static analysis of the MDOF model of the building was first conducted to calculate the flexible and stiff-edge 3D displacements and 2D displacements in each storey '$i$': $\delta_{flexible,i}$, $\delta_{stiff,i}$, and $\delta_{2D,i}$, respectively. The torsional parameters: $e_r$, $b_r$, and $T_n$, were then calculated using Equations (A1a)–(A1f):

$$\Delta_{2D} = \frac{\sum_{i=1}^{n} m_i \delta_{2D,i}^2}{\sum_{i=1}^{n} m_i \delta_{2D,i}} \tag{A1a}$$

$$\Delta_{max} = \frac{\sum_{i=1}^{n} m_i \delta_{flexible,i}^2}{\sum_{i=1}^{n} m_i \delta_{flexible,i}} \tag{A1b}$$

$$\Delta_{min} = \frac{\sum_{i=1}^{n} m_i \delta_{stiff,i}^2}{\sum_{i=1}^{n} m_i \delta_{stiff,i}} \tag{A1c}$$

$$CR_{from\ the\ stiff\ edge} = \frac{(\Delta_{2D} - \Delta_{min})L}{\Delta_{max} - \Delta_{min}} \tag{A1d}$$

$$e = B - CR_{from\ the\ stiff\ edge},\ e_r = \frac{e}{r},\ T_n = 2\pi \sqrt{\frac{\sum_{i=1}^{n} m_i \delta_i}{e_r}} \tag{A1e}$$

$$b_r = \frac{1}{r}\sqrt{\frac{\Delta_{2D} L e_s}{\Delta_{max} - \Delta_{min}}},\ r = \sqrt{\frac{L_x^2 + L_y^2}{12}}\ \text{for rectangular floor plan} \tag{A1f}$$

where $n$ is the number of stories of a multi-storey building, $m_i$ is the mass for storey '$i$', $\Delta_{2D}$ is the translational displacement for the SDOF model, and $\Delta_{max}$ and $\Delta_{min}$ are the displacements for the flexible and stiff edges of the single-storey model. $V_b$ is the total base shear calculated as per the relevant seismic code, L is the furthest distance of the edge elements, $L_x$ and $L_y$ are the length and width of the rectangular floor plan, $B$ is the distance of the stiff edge from the centre of mass, and $e_s$ is the distance of applied load from the centre of mass.

## Appendix C. Implementation of Newmark Constant Average Acceleration Time Step Integration in Rapid Nonlinear Time History Analysis in 2D (RNLTHA-2D)

The linear displacement '$u(t + \Delta t)_{linear}$' and the nonlinear displacement '$u(t + \Delta t)_{NL}$' responses at the current time step are determined from Equations (A2) and (A3), respectively:

$$u(t + \Delta t)_{linear} = \frac{\left(-\ddot{u}_g(t + \Delta t) + \left(\frac{2}{\Delta t^2} + \frac{4\xi w_n}{\Delta t}\right)u(t) + \left(\frac{4}{\Delta t} + 2\xi w_n\right)\dot{u}(t) + \ddot{u}(t)\right)}{\left(\frac{2}{\Delta t^2} + \frac{4\xi w_n}{\Delta t} + w_n^2\right)} \tag{A2}$$

$$u(t + \Delta t)_{NL} = u(t) + \frac{\left(-\ddot{u}_g(t + \Delta t) + \left(\frac{2}{\Delta t^2} + \frac{4\zeta w_s}{\Delta t}\right)u(t) + \left(\frac{4}{\Delta t} + 2\zeta w_s\right)\dot{u}(t) + \ddot{u}(t)\right) - F_N/M_j}{\left(\frac{2}{\Delta t^2} + \frac{4\zeta w_s}{\Delta t} + w_s^2\right)} \tag{A3}$$

where $u(t)$, $\dot{u}(t)$, and $\ddot{u}(t)$ are the displacement, velocity, and acceleration at the previous time step '$t$', $\ddot{u}_g(t)$ is the ground acceleration due to seismic excitation at the current time step '$t + \Delta t$', $\Delta t$ is the time step, $w_n$ and $w_s$ are the angular velocity based on the initial and secant stiffness, $\zeta$ is the damping ratio, $F_N$ is the nonlinear force, and $M_j$ is the modal mass.

## Appendix D. Trilinear Hysteresis Model

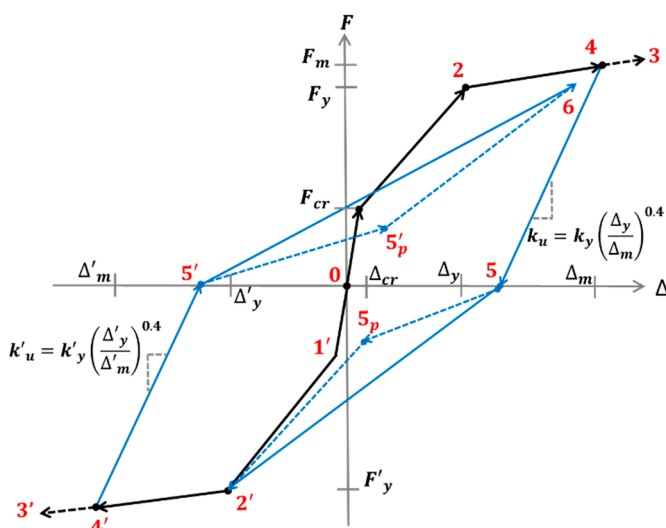

**Figure A2.** Trilinear hysteresis model [23,24]. The dotted blue line represents the option to include pinching.

## Appendix E. Material and Dynamic Properties of the Case Study Building

**Table A1.** The material properties of RC walls of the case study building.

| Parameters | Walls 1 and 2 |
|---|---|
| Diameter of vertical reinforcement ($d_v$) | 20 mm |
| Vertical reinforcement ratio ($p_v$) | 0.015 (1.5%) |
| Yield strength of reinforcement ($f_{sy}$) | 500 MPa |
| Ultimate strength of reinforcement ($f_{su}$) | 600 MPa |
| Characteristic strength of concrete ($f_c'$) | 40 MPa |
| Axial load ratio (n) | 0.11 |

**Table A2.** Dynamic properties of the case study building obtained from SeismoStruct [41].

| Parameters | Mode 1 | Mode 2 | Mode 3 |
|---|---|---|---|
| Effective mass (tonnes) | 2735 | 840 | 290 |
| Mass participation ratio (%) | 65 | 20 | 7 |
| Period (s) | 0.810 | 0.129 | 0.046 |
| Displacement coefficient at the roof level, $P_{j,roof}$ | 1.56 | 0.70 | 0.33 |

## Appendix F. Details of the Site−Specific Accelerograms Generated from QuakeAdvice

**Table A3.** Summary of the 24 earthquake records selected from the PEER database.

| Spectra No. | Acc No. | Earthquake Name | Reference Periods (s) | Year | Station Name | Magnitude | $R_{jb}$ (km) | PGA (g) | Scaling Factor |
|---|---|---|---|---|---|---|---|---|---|
| 1 | 1 | Whittier Narrows-02 | 0.2 | 1987 | Mt Wilson—CIT Seis Sta | 5.27 | 16.45 | 0.175 | 1.21 |
| 2 | 2 | Northridge-06 | 0.2 | 1994 | Beverly Hills—12520 Mulhol | 5.28 | 10.57 | 0.130 | 0.85 |

**Table A3.** *Cont.*

| Spectra No. | Acc No. | Earthquake Name | Reference Periods (s) | Year | Station Name | Magnitude | $R_{jb}$ (km) | PGA (g) | Scaling Factor |
|---|---|---|---|---|---|---|---|---|---|
| 3 | | Christchurch—2011 | 0.2 | 2011 | PARS | 5.79 | 8.5 | 0.126 | 0.61 |
| 4 | | Sierra Madre | 0.2 | 1991 | Cogswell Dam—Right Abutment | 5.61 | 17.79 | 0.151 | 0.50 |
| 5 | | Friuli (aftershock 9)_ Italy | 0.2 | 1976 | San Rocco | 5.5 | 11.92 | 0.127 | 1.41 |
| 6 | | Lytle Creek | 0.2 | 1970 | Wrightwood—6074 Park Dr | 5.33 | 10.7 | 0.215 | 1.06 |
| 7 | 3 | Christchurch—2011 | 0.5 | 2011 | GODS | 5.79 | 9.1 | 0.175 | 0.63 |
| 8 | 4 | Chi-Chi_ Taiwan-05 | 0.5 | 1999 | HWA031 | 6.2 | 39.29 | 0.128 | 1.91 |
| 9 | 5 | Chi-Chi_ Taiwan-05 | 0.5 | 1999 | HWA005 | 6.2 | 32.71 | 0.124 | 1.46 |
| 10 | 6 | Whittier Narrows-01 | 0.5 | 1987 | Pacoima Kagel Canyon | 5.99 | 31.59 | 0.169 | 1.04 |
| 11 | | Chi-Chi_ Taiwan-03 | 0.5 | 1999 | CHY041 | 6.2 | 40.79 | 0.132 | 1.00 |
| 12 | | N. Palm Springs | 0.5 | 1986 | Anza—Red Mountain | 6.06 | 38.22 | 0.171 | 1.77 |
| 13 | 7 | Chi-Chi_ Taiwan-06 | 1 | 1999 | CHY041 | 6.3 | 45.68 | 0.094 | 0.53 |
| 14 | 8 | San Fernando | 1 | 1971 | Lake Hughes #4 | 6.61 | 19.45 | 0.198 | 1.27 |
| 15 | 9 | Coalinga-01 | 1 | 1983 | Parkfield—Fault Zone 11 | 6.36 | 27.1 | 0.084 | 1.08 |
| 16 | 10 | Coalinga-01 | 1 | 1983 | Parkfield—Stone Corral 3E | 6.36 | 32.81 | 0.170 | 1.13 |
| 17 | 11 | Northridge-01 | 1 | 1994 | LA—Temple & Hope | 6.69 | 28.82 | 0.113 | 0.62 |
| 18 | 12 | Niigata_ Japan | 1 | 2004 | NGNH29 | 6.63 | 45.39 | 0.193 | 1.58 |
| 19 | 13 | Loma Prieta | 2 | 1989 | SF—Diamond Heights | 6.93 | 71.23 | 0.076 | 0.67 |
| 20 | 14 | Iwate_ Japan | 2 | 2008 | Maekawa Miyagi Kawasaki City | 6.9 | 74.82 | 0.159 | 0.95 |
| 21 | | Chuetsu-oki_ Japan | 2 | 2007 | NGNH28 | 6.8 | 76.68 | 0.051 | 1.42 |
| 22 | | Iwate_ Japan | 2 | 2008 | AKT009 | 6.9 | 118.9 | 0.086 | 1.66 |
| 23 | | Loma Prieta | 2 | 1989 | Berkeley—Strawberry Canyon | 6.93 | 78.32 | 0.077 | 1.01 |
| 24 | | Chuetsu-oki_ Japan | 2 | 2007 | NGNH27 | 6.8 | 91.38 | 0.050 | 1.29 |

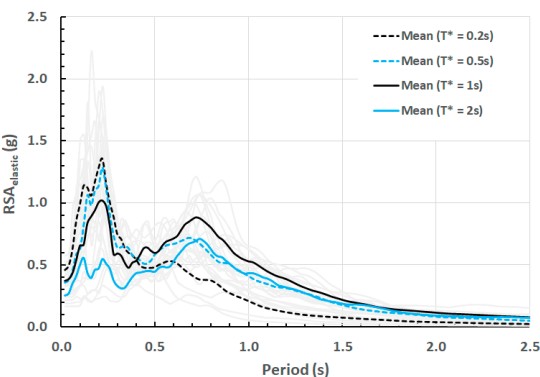

**Figure A3.** Site-specific mean acceleration response spectra of the four reference periods.

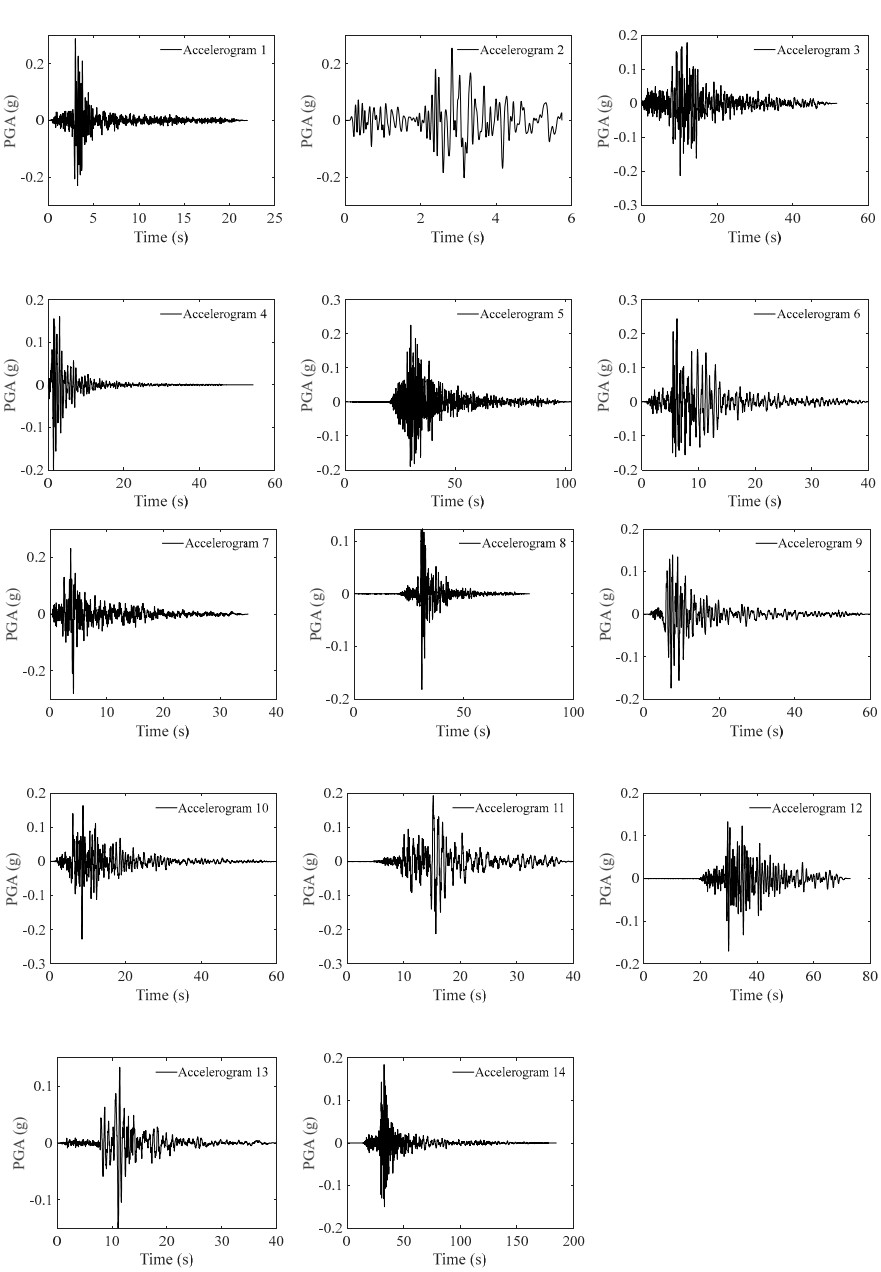

**Figure A4.** Soil−surface accelerograms generated from QuakeAdvice.

## Appendix G. Comparison of the 3D Nonlinear Displacement Time History Response of the Case Study Building

**Table A4.** Comparison of the maximum 3D roof displacement obtained from the proposed rapid nonlinear time history analysis in 3D (RNLTHA-3D) and SeismoStruct.

| No. | $\Delta_{3D}/\Delta_{2D}$ | $\Delta_{roof,max}$ (mm) RNLTHA-2D | $\Delta_{roof,max}$ (mm) RNLTHA-3D | $\Delta_{roof,max}$ (mm) SeismoStruct-3D |
|---|---|---|---|---|
| 1 | 1.14 | 27 | 31 | 39 |
| 2 | 1.11 | 26 | 29 | 31 |
| 3 | 1.87 | 109 | 204 | 182 |
| 4 | 1.86 | 75 | 140 | 161 |
| 5 | 1.38 | 112 | 155 | 194 |
| 6 | 1.69 | 90 | 152 | 127 |
| 7 | 2.39 | 132 | 315 | 303 |
| 8 | 1.87 | 140 | 262 | 226 |
| 9 | 1.84 | 167 | 307 | 262 |
| 10 | 1.61 | 95 | 153 | 163 |
| 11 | 1.54 | 182 | 280 | 263 |
| 12 | 1.67 | 106 | 177 | 190 |
| 13 | 1.71 | 65 | 111 | 135 |
| 14 | 1.78 | 82 | 146 | 168 |

## Appendix H. Comparison of the 3D Nonlinear Displacement Time History Response of the Case Study Building #2

To make the results more convincing, one more case studying building #2 that has a different configuration of structural walls has also been investigated, as shown in Figure A5. The comparison of the maximum deflection for case study building #2 estimated by the rapid nonlinear time history analysis in 3D (RNLTHA-3D) against predictions by SeismoStruct has been extended to cover for earthquake excitations corresponding to the four reference periods, as shown in Figure A6 and Table A5. Comparisons of the peak displacement for each accelerogram are listed in Table A6. The comparison statistics show overall good agreement, validating the effectiveness of the proposed methodology in predicting seismic responses of buildings in 3D.

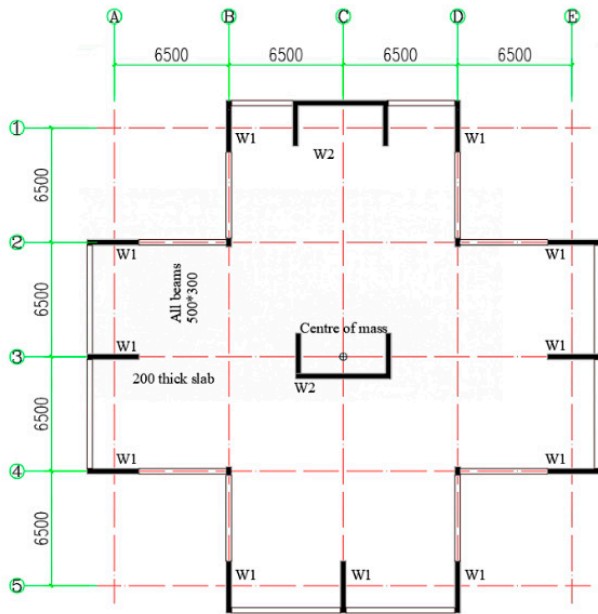

**Figure A5.** Floor plan of case study building #2.

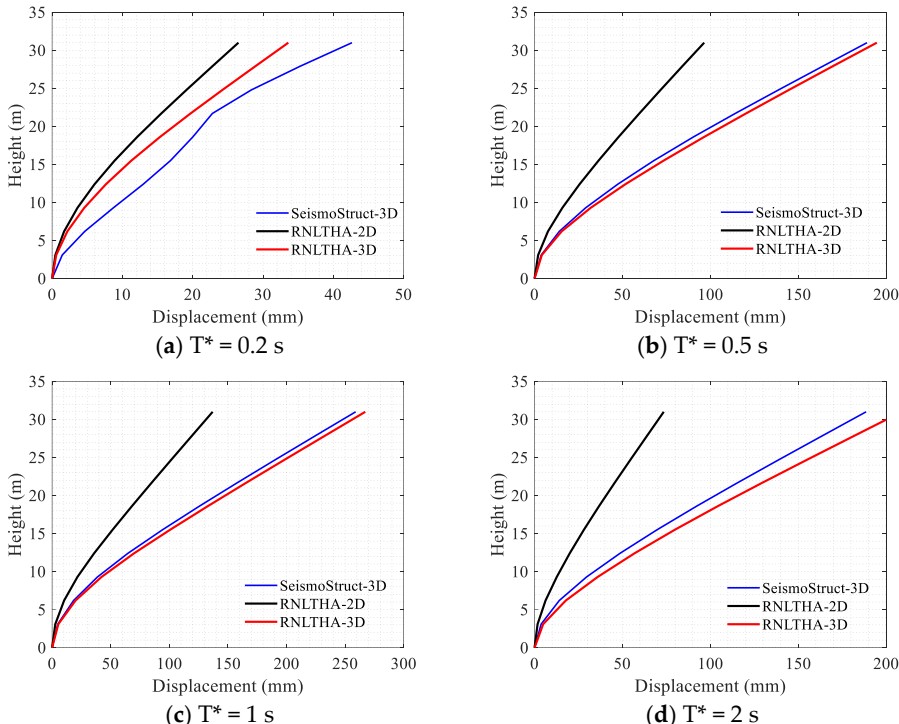

**Figure A6.** Comparison of the maximum displacement profile of case study building #2 obtained using RNLTHA-3D and SeismoStruct.

**Table A5.** Comparison of maximum 3D roof displacement of case study building #2 obtained from the proposed RNLTHA-3D and SeismoStruct for the four reference periods.

| Reference Period (T*) | $\Delta_{roof,\text{RNLTHA}-\text{3D}}$ (mm) | $\Delta_{roof,\text{SeismoStruct}}$ (mm) | Difference $\left(\frac{\Delta_{roof,\text{SeismoStruct}}-\Delta_{roof,\text{RNLTHA}-\text{3D}}}{\Delta_{roof,\text{SeismoStruct}}}\right)\times 100\%$ |
|---|---|---|---|
| 0.2 s | 34 | 42 | −19.0% |
| 0.5 s | 191 | 189 | 1.0% |
| 1 s | 257 | 259 | −0.6% |
| 2 s | 204 | 189 | 8.1% |

**Table A6.** Comparison of maximum 3D roof displacement of case study building #2 obtained from RNLTHA-3D and SeismoStruct for each accelerogram.

| Earthquake No. | $\Delta_{3D}/\Delta_{2D}$ | $\Delta_{roof,max}$ (mm) RNLTHA-2D | $\Delta_{roof,max}$ (mm) RNLTHA-3D | $\Delta_{roof,max}$ (mm) SeismoStruct-3D |
|---|---|---|---|---|
| 1 | 1.15 | 27 | 31 | 49 |
| 2 | 1.38 | 26 | 36 | 36 |
| 3 | 1.98 | 109 | 216 | 209 |
| 4 | 2.29 | 75 | 171 | 187 |
| 5 | 1.58 | 112 | 177 | 165 |
| 6 | 2.23 | 90 | 200 | 196 |
| 7 | 2.03 | 132 | 268 | 279 |
| 8 | 1.54 | 140 | 215 | 241 |
| 9 | 1.85 | 167 | 308 | 305 |
| 10 | 2.27 | 95 | 216 | 236 |
| 11 | 1.46 | 182 | 265 | 243 |
| 12 | 2.56 | 106 | 272 | 250 |
| 13 | 3.43 | 65 | 223 | 203 |
| 14 | 2.25 | 82 | 185 | 175 |

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
