# Peer review of "Assessment of Torsional Amplification of Drift Demand in a Building Employing Site-Specific Response Spectra and Accelerograms"

_2673-4109, doi:10.3390/civileng4010015_

Round 1
Reviewer 1 Report
The proposed modeling technique is practical for design engineer to apply in industrial project. However, in this paper, the case study focused on the analysis of a 10-storey RC building only. More types of buildings should be studied (eg. height and core wall position that may affect the torsional stiffness of the building) to validate the proposed modeling technique and expand the content of the paper.
Reviewer 2 Report
This paper is generally well-written. Research is adequate. However, some clarification may be required for the benefit of normal readers, as noted below. Authors can address with minor revisions.
1. For example, Fig 8a - Δ3?/Δ2? obtained from Routine 1, shows values up to 130 mm, whereas 8b, shows values scaled up to 200 mm. Need to state the multiplication factor for the full range (even if a constant value was used).
2. The captions say the comparison is between RNLTHA-2D and SeismoStruct whereas comparisons with 3D is also presented. Therefore, the caption needs to be revised.
3. In all cases in Fig 9 the 3D displacement demand is more than 2D. Therefore, the 2D models underestimate the DD?
4. Based on fig 9, it can also be stated that starting from a T*=0.5 s and above, this method is more suited.
5. I didn't have the chance to read the companion papers. However, the methodologies presented require a certain level of expertise for a practitioner to understand and use these methods. It can be stated that the research addresses the limitations in current practice and offers reasonable solutions.
Round 2
Reviewer 1 Report
Thanks for the revision. There is no further comment.